# The Effect of Ascorbic Acid on Hepatic Ischaemia–Reperfusion Injury in Wistar Rats: An Experimental Study

**DOI:** 10.3390/ijms25168833

**Published:** 2024-08-14

**Authors:** Jorge Luiz Saraiva Ximenes, Joel Avancini Rocha-Filho, Flavio Henrique Ferreira Galvão, Cinthia Lanchotte, Marcia Saldanha Kubrusly, Regina Maria Cubero Leitão, Jose Jukemura, Agustin Vintimilla Moscoso, Emilio Elias Abdo, Luiz Augusto Carneiro D’Albuquerque, Estela Regina Ramos Figueira

**Affiliations:** 1Laboratório de Investigação Medica 37, Departamento de Gastroenterologia, Hospital das Clínicas HCFMUSP, Faculdade de Medicina, Universidade de São Paulo, São Paulo 05508-220, SP, Brazil; jlsximenes@gmail.com (J.L.S.X.); joelrocha@me.com (J.A.R.-F.); flavio.galvao@hc.fm.usp.br (F.H.F.G.); cinthia.lanchotte@fm.usp.br (C.L.); msk@alumni.usp.br (M.S.K.); reginabg10@gmail.com (R.M.C.L.); j.jukemura@hc.fm.usp.br (J.J.); abdow@alumni.usp.br (E.E.A.); profluizcarneiro@gmail.com (L.A.C.D.); 2Disciplina de Anestesiologia, Hospital das Clínicas HCFMUSP, Faculdade de Medicina, Universidade de São Paulo, São Paulo 05508-220, SP, Brazil; 3Serviço de Transplante de Fígado e Órgãos do Aparelho Digestivo, Departamento de Gastroenterologia, Hospital das Clínicas HCFMUSP, Faculdade de Medicina, Universidade de São Paulo, São Paulo 05508-220, SP, Brazil; 4Divisão de Cirurgia do Aparelho Digestivo, Departamento de Gastroenterologia, Hospital das Clínicas HCFMUSP, Faculdade de Medicina, Universidade de São Paulo, São Paulo 05508-220, SP, Brazil; 5Departamento de Trasplantes do Hospital “José Carrasco Arteaga” IESS, Cuenca 010109, Ecuador; dr.agustinvintimilla@gmail.com

**Keywords:** ischaemia, reperfusion, ascorbic acid, liver, experimental

## Abstract

Liver ischaemia–reperfusion (IR) during hepatic surgeries can lead to liver cell death via oxidative stress and the activation of immune cells, the release of cytokines, and damage-associated molecular patterns. Ascorbic acid has been shown to confer potential protective effects against IR injury, mainly due to its antioxidant properties. This study evaluated the effect of ascorbic acid infusion at different time points during hepatic IR in rats. Thirty-six male Wistar rats were divided into control and experimental groups that received the same total ascorbic acid dose at three different infusion times: before ischaemia, before reperfusion, or before both ischaemia and reperfusion. All of the animals experienced hepatic IR injury. We measured the hepatic enzymes, cytokines, and portal blood flow. Animals receiving ascorbic acid before both ischaemia and reperfusion had lower liver enzyme levels, reduced inflammation, and better portal venous flow than other animals. Divided doses of ascorbic acid before IR may be beneficial for reducing liver injury associated with IR.

## 1. Introduction

Liver ischaemia–reperfusion (IR) has long been a problem associated with hepatic surgery, including resections and transplants [1,2,3,4,5,6,7,8,9]. The issue derives from the massive production of free oxygen and nitrogen radicals; the activation of Kupfer cells, neutrophils, and lymphocytes; the expression of damage-associated molecular patterns; the production and secretion of cytokines; and the activation of TOLL-like receptors (TLRs) and proteins of the high-mobility group box-1 (HMGB-1). All of these processes can lead to necrosis and apoptosis of the hepatic parenchyma [1,2,3,7,10,11,12,13,14,15]. 

Ascorbic acid (AA), or vitamin C, is a natural antioxidant acquired via nutrition; its venous infusion has been shown to have a potential protective effect against IR injury in experimental studies [15,16,17,18,19,20,21,22,23,24]. It is a soluble, low-cost, and easily administrated agent that can reduce free radicals and potentially act to modulate the inflammatory cascade, confer mitochondrial and endothelial protection, reduce lipid peroxidation, limit membrane and nuclear damage, and limit the elevation of transaminases after liver IR [16,17,19,25,26,27,28]. Ascorbic acid can be administered before ischaemia, before reperfusion, or at both time points; the best administration protocol remains unknown. 

We evaluated the effect of AA infusion at three different time points for hepatic IR injury in an experimental rat model.

## 2. Results

Of the 36 animals, 27 were used in our statistical analyses. One animal in the control group died during anaesthesia; the laboratory analysis failed in eight animals because of sample coagulation or inconclusive blood gas analysis results (three animals in the control group, one animal in the PI group, one animal in the PR group, and three animals in the PIPR group). The C (control) group, therefore, contained five animals; the PIPR (pre-ischaemia and pre-reperfusion) group contained six animals; and the PR (pre-reperfusion) and PI (pre-ischaemia) groups both contained eight animals each.

### 2.1. Biochemical Results

The transaminase dosages were less elevated in ALT in the PIPR group (1227 ± 420 UI/L) than in the C (2300 ± 686 UI/L), *p* = 0.005 and PR groups (3001 ± 1576 UI/L), *p* = 0.01. A similar effect was not observed when we compared PIPR with the PI group (2341 ± 1907 UI/L), *p* > 0.05. Furthermore, there were no significant changes in the other comparisons (*p* > 0.05). The AST dosage was also lower in the PIPR group (1715 ± 576 UI/L) than in the C (2541 ± 677 UI/L) (*p* = 0.028) and PR groups (3874 ± 2388 UI/L) (*p* = 0.026); there were no significant changes in comparison with the PI group (2437 ± 1412 UI/L) and the other comparisons (*p* > 0.05). The data are shown in Figure 1 and Figure 2.

The blood gas analysis data are summarised in Table 1. The pH and BIC measurements were the highest in the PI group; the base excess (BE) dosage was less negative, and lactate was also lower in this group than that in the other groups. The differences were statistically significant for BIC and BE (*p* = 0.027 and *p* = 0.035, respectively). The other differences were not statistically significant, however (*p* > 0.05). The potassium dosage was lower in the control group than that in the PR and PI groups (*p* = 0.047 and *p* = 0.048, respectively). However, the differences were not statistically significant when compared with the PIPR group. Calcium dosages were lower in the PR and PIPR groups that in the other groups, but the differences were not statistically significant (*p* > 0.05). The serum glucose measurements after 4 h of reperfusion did not exhibit any statistically significant differences among the groups. The haemoglobin dosage was significantly lower in the PIPR group than that in the control group (*p* = 0.01), but there were no significant changes in any of the other comparisons (*p* > 0.05).

In terms of interleukin dosages, we noted lower levels of IL-1beta in the PI group than in the PIPR group (*p* = 0.036) and also lower levels of IL-10 in the PI group than in the PR and PIPR groups (*p* = 0.039 and *p* = 0.045, respectively). However, there were no statistically significant differences between the other groups (*p* > 0.05). The IL-1beta values in the PI and PIPR groups were lower than those in the C group. The IL-6 and IL-10 values were also higher in the PR group, but the difference was not statistically significant. The differences in the IL-12 and TNF-a values were also not statistically significant. Our HMGB-1 analyses revealed lower levels in the PIPR group, and the difference compared to the PR group was statistically significant (*p* = 0.034). However, none of the other comparisons attained statistical significance (*p* > 0.05). These data are summarised in Figure 3 and Table 2.

### 2.2. Portal Venous Flow

The two-way ANOVA of the portal venous flow measures revealed statistically significant differences on the basis of administration time (*p* < 0.01) but not between the groups (*p* > 0.05). Using the unpaired Student’s *t*-test for two-by-two analysis, we found that the portal flow values measured before ischaemia (baseline portal flow) did not differ between groups (*p* > 0.05). The portal flow measurements obtained after 5 min of reperfusion were higher in the PIPR group; the differences between the PI and PR groups were statistically significant (*p* = 0.01 and *p* = 0.02, respectively). There was also a significant difference between the C group and the PI group (*p* = 0.03). The portal flow values after 4 h of reperfusion did not differ between the groups (*p* > 0.05). 

When we focused on the portal flows at each time period within the same group, we noted a significant reduction in portal flow after 5 min of reperfusion compared with the baseline value (*p* < 0.01 in the C, PI, and PR groups; *p* = 0.02 in the PIPR group). Animals in the PI and PR groups exhibited a significant increase in portal flow after 4 h of reperfusion compared with after 5 min (*p* = 0.01); the animals in the C and PIPR groups did not exhibit a significant increase (*p* = 0.26 and *p* = 0.42, respectively). After 4 h of reperfusion, the animals in all of the groups exhibited significantly lower portal flow than the baseline values (*p* < 0.01 for the C, PI, and PR groups; *p* = 0.02 for the PIPR group). These data are summarised in Figure 4.

### 2.3. Histopathology

Our histopathological analysis revealed reduced necrosis and congestion in the animals in the PIPR group compared with the animals in the PI and PR groups. However, the differences were not statistically significant. Additionally, there were no statistically significant differences between the groups regarding the degrees of steatosis, portal and lobular inflammatory infiltrate, sinusoidal cells, or detrabeculation. The histological score (adapted from Quireze et al. [29]) was not statistically significantly different between the groups (*p* > 0.05). The descriptive data are summarised in Table 3. Examples of the histopathological changes are shown in Figure 5.

## 3. Discussion

Oxidative stress is involved in several chronic diseases and in post-IR injuries. Reactive oxygen species (ROS) are mainly produced in the mitochondria, and an excess of such species, which can persist after reperfusions, can alter the redox state of an organ or tissue and promote inflammation and necrosis via the dysregulation of signals and cytokines and direct damage to lipids, proteins, and carbohydrates [11,14,15,29,30]. In liver surgeries, frequent clamping in the region of the hepatic hilum (i.e., Pringle’s manoeuvre [9]) can cause periods of normothermic ischaemia followed by reperfusions, with local and systemic repercussions [9,31,32]. In addition, a period of hours of cold ischaemia in liver transplants is combined with periods of warm ischaemia, with possible repercussions on the functioning of the graft [33,34]. Several forms of pre- or post-conditioning have been tested to reduce IR injury; these interventions have included volatile anaesthetics [35] and new experimental drugs [36,37,38]. 

Ascorbic acid itself may be a viable solution, however. Infusions of AA in a peripheral vein appear to protect the liver tissue from IR injuries. We showed in male Wistar rats that experienced ischaemia of the median and left anterolateral hepatic lobes and subsequent removal of the non-ischaemic right (upper and lower) and caudate (lower and upper) lobes that there was a lower mean dosage of AST and ALT in animals that received AA. The effect was greater when the infusion was divided (i.e., half of the dose before ischaemia and half before reperfusion). A similar result regarding the timing of the administration of AA was observed by Figueira et al. [35]. Those authors tested the efficacy of sevoflurane for attenuating hepatic IR. The administration of sevoflurane in pre-conditioning (before ischaemia) and the combination of pre-conditioning and post-conditioning (before reperfusion) attenuated the LIR and acid–base imbalance. However, only the group that underwent pre- and post-conditioning exhibited increased haemodynamic recovery [35].

High doses of AA can maintain plasma concentrations for a long time during the most oxidative phase of reperfusion. Critically ill patients who receive a 1 g bolus (approximately 10–15 mg/kg for an average patient weighing 70 kg) exhibit high plasmatic concentrations (40 mg/L or 0.25 mMol/L) in the first hour. That level then drops to within the normal range within 12 h [39]. On the other hand, 5 g boluses (approximately 71 mg/kg for an average patient weighing 70 kg) result in supranormal plasma concentrations of approximately 200 mg/L (approximately 1 mMol/L). Despite falling, those levels remain above normal for 12 h [40]. It is worth remembering that, when preventing hepatic IR injury, the objective is to maintain optimal doses in the hepatic tissue; AA is rapidly consumed during oxidative states [39,41].

Extracellular levels of AA in the hepatic tissues increase during ischaemia and during the beginning of reperfusion; a reduction then ensues in the subsequent minutes [42]. That situation indicates the displacement of AA to the ischaemic tissues and its ultimate consumption. The levels of AA in ischaemic hepatic lobes are also lower than those in non-ischaemic hepatic lobes [29]. This finding is likely due to the rapid consumption of AA. An infusion of AA before ischaemia sets up a stockpile, and a second dose before reperfusion can optimise the treatment. That second dose will have a more pronounced effect on reperfusion, and the displacement of AA from the rest of the body to the affected area can begin.

The initial stages of IR-injury-attenuating strategies can affect their ultimate effects; differences between pre-ischaemia and pre-reperfusion administration have already been observed [35,36,38,43]. Here, we highlight the utility of administering a divided dose of AA: half before ischaemia and half before reperfusion. This situation can be observed in the expression of liver damage by transaminases; animals in the PIPR group—the group that received half a dose of AA before ischaemia and the other half before reperfusion—exhibited the best profiles. Ascorbic acid is very water soluble; after parenteral infusion, it is distributed throughout tissues and has an immediate effect [24,25,28,44,45,46]. This infusion strategy could be utilised during anaesthesia for both liver resections and transplants: in resections, one dose could be given at the start of surgery and another after periods of ischaemia; for transplants, doses could be administered before organ procurement and again prior to reperfusion.

Hepatic IR can result in several hemodynamic changes [32], some of which have been studied previously by our group [35,36]. Clamping of the hepatic hilum may reduce the cardiac pre-load and directly affect the hemodynamic balance and, after reperfusion, inflammatory mediators may affect the vascular tone. Our group evaluated the portal venous flow and found a consistent reduction after IR in all groups, as observed previously [36]. The “no reflow” phenomenon, in which vascular flow does not fully recover after a period of ischaemia, frequently occurs after hepatic ischaemia [32]. The intravenous administration of AA may play a role in optimising the recovery of the portal venous flow after a late IR; however, this topic should be more thoroughly analysed in future studies.

The primary mechanism of action of AA is its antioxidant effect. That effect was evidenced in our work by the increased acid–basic equilibrium in the animals in the groups that received AA before ischaemia. However, there are other possible benefits of AA. Toll-like receptors and the proteins of the HMGB-1 are other targets [47,48,49]. These proteins are expressed in various tissues, and they are largely stored in the cell nuclei and participate in DNA replication, recombination, translocation, and repair; they also interact with other transcription factors such as p53 and NF-Kb and play a complex role in both cell regeneration and inflammation [30,50,51,52]. They have three domains with cysteine that are sensitive to the redox state. In an oxidative environment, they act to promote inflammation [30] by binding to TLRs and activating the release of cytokines by macrophages [47,48,53,54]. Those effects have been observed in hepatic, cerebral, renal, skeletal, and cardiac muscle IR in experimental models [30,48,55,56,57,58]. The presence of HMGB-1 in ischaemia and reperfusion has been studied as a potential marker for the functionality of liver grafts and transplants [59,60,61]. 

The reduction in hepatic IR injury promoted by a metabolite of sevoflurane may be conferred via the inhibition of HMGB-1 [36]. In this study, we noted lower levels of HMGB-1 in the PIPR group. Another important factor is the regulation of HMGB-1; the activity of these proteins can induce tolerance or inflammation depending on the redox state of the tissue in which they are released [30]. Given that AA acts as an antioxidant and replenishes scavenger mechanisms capable of maintaining a better redox state [23,24,25,28,44,45,46], a high concentration prior to ischaemia and reperfusion can preserve this regulation and assist in preventing liver injury after an IR insult when paired with the reduced extracellular release of HMGB-1. That situation can potentially improve the action of AA in subsequent tissue recovery.

This study has some limitations, even so, it provides useful information about the profile of liver injury and the metabolism of IR and its attenuation via AA at different dosages and times. The use of secondary endpoints, such as transaminases and interleukins and HMGB-1, has limitations; they do not always translate into real clinical benefits, but they can indicate a path forward. In many liver surgeries, several short periods of warm ischaemia often occur that can exceed 30–40 min in total. In liver transplants, cold ischaemia can occur that lasts for hours. At that time, necrosis may not be evident, despite the presence of hepatocytes lesions. That situation was observed in our study; the release of transaminases was a good indicator of hepatic aggression.

A more complete assessment of the effects of AA would require a study that observed the markers of liver function and the survival time and was characterised by a longer follow-up duration. The time of reperfusion might have also influenced the markers; longer times might yield different results, such as more pronounced histopathological changes than those seen in this study. In addition, AA has a large safety margin and dosage variation; higher doses can trigger different results for different durations of IR. That situation can make it difficult to fully explore the potential of AA in one study. Therefore, it would be interesting to carry out more studies characterised by different durations of IR and AA doses. Larger samples and greater powers would also be useful for better defining the role that AA plays in protecting against IRI.

In conclusion, an infusion of AA in a divided dose before ischaemia and before reperfusion was particularly effective at reducing transaminases and inflammation; increased metabolic recovery after hepatic IRI may help to improve portal venous flow recovery.

## 4. Materials and Methods

### 4.1. Animals and Study Design

This study was approved by the University of Sao Paulo School of Medicine’s Ethics Committee on the Use of Animals (No. 1436/2020). A total of 36 male Wistar rats (*Rattus novergicus albinus*) weighing between 250 and 300 g were anaesthetised and exposed to 30 min of hepatic ischaemia and 4 h of reperfusion. The rats were then randomised into four groups of eight animals each: a control (C) group received an intravenous saline solution; a pre-ischaemia (PI) group received a full dose of AA at 100 mg/kg 10 min before ischaemia; a pre-reperfusion (PR) group received a full dose of AA 10 min before reperfusion; and a pre-ischaemia and pre-reperfusion (PIPR) group received half a dose of AA (i.e., 50 mg/kg) 10 min before ischaemia and the other half 10 min before reperfusion. The experimental design of the study is outlined in Figure 6. The selected dose was determined based on existing pharmacokinetic data [40] and prior animal studies [16,17,19,20].

### 4.2. Anaesthesia and Surgical Procedure

All of the animals were housed in individual cages with free access to food and water, at a controlled temperature held between 20 and 23 degrees Celsius. They underwent the same anaesthetic and surgical procedures performed in other studies of our group [36]. The animals were anaesthetised intraperitoneally using ketamine hydrochloride 5% (Ketalar^®^, Cristália, São Paulo, SP, Brazil), 60–80 mg/kg, and xylazine hydrochloride 2% (Rompum^®^, Bayer, SP, Brazil), 5–7 mg/kg. They underwent standardised orotracheal intubation with Jelco^®^ 18G and mechanical ventilation (Small Animal Ventilator model 683; Harvard Apparatus, Holliston, MA, USA) with a tidal volume of 0.08 mL/g of body weight, a respiratory rate of 60/min, and a 21% fraction of inspired oxygen (FiO_2_) [62]. A median laparotomy extending for about 4 cm from the xiphoid appendix was performed. The portal vein was isolated to measure the basal portal flow. We also dissected the common pedicle of the median and left anterolateral lobes, which was occluded using an atraumatic microvascular clamp (MiniClamp Bulldog dietrich, Prime Instrumentos Cirúrgicos, São Paulo, SP, Brazil). That process enabled specific ischaemia of 70–80% of the liver volume without obstructing the splanchnic flow. After 30 min, the clamp was removed, reperfusion was initiated, and the non-ischaemic right and caudate lobes (upper and lower portions) were immediately resected, and the portal venous flow was recorded. The abdominal incision was closed with continuous sutures, and the analgesia was supplemented with morphine 5 mg/kg subcutaneously. Each rat was then allowed to recover in an individual cage. After 4 h, the animals were re-anaesthetised to measure the portal venous flow and obtain blood and tissue samples. Finally, the animals were euthanised via exsanguination.

### 4.3. Portal Venous Flow

The portal venous flow was measured using a perivascular probe (PROBE NAME, Transonic Systems Inc., Ithaca, NY, USA) connected to a flow meter (TS420 Animal Research Flowmeter, Transonic Systems Inc., Ithaca, NY, USA). We conducted those measurements three times during the procedure: before starting ischaemia (basal portal flow), 5 min after reperfusion, and 4 h after reperfusion.

### 4.4. Laboratory Analyses (Liver Enzymes, Blood Gas, Interleukins, and High-Mobility Group Box-1)

We evaluated transaminases, aspartate aminotransferase (AST), and alanine aminotransferase (ALT) 4 h after reperfusion. We quantified AST and ALT (in IU/L) using the optimised ultraviolet method (COBAS MIRA, Roche Diagnostics, Rotkreuz, Switzerland). The results were expressed in units per litre. Glucose (mg/dL), lactate (mg/dL), potassium (mmol/L), calcium (mg/dL), pH, bicarbonate (BIC) (mmol/L), and haemoglobin (g/dL) were quantified using a gas analyser (ABL800 Flex; Radiometer Medical ApS, Brønshøj, Denmark).

We quantified TNF-α, interleukins IL-6 and IL-10, and HMGB-1 using an immunoenzymatic assay (ELISA, Bio-Rad, Hercules, CA, USA) in plasma samples 4 h after the start of liver reperfusion. The results were expressed in picograms per mL.

### 4.5. Histopathological Data Collection and Analyses

A histological analysis was performed on liver samples obtained 4 h after the start of reperfusion. Fragments of liver tissue, fixed in 2% formalin solution and stained with haematoxylin and eosin, were histologically analysed using optical microscopy. A single pathologist, blinded to the study groups, performed the analyses. We analysed the following parameters: congestion (cellular enlargement linked to the entry of intracellular water), portal inflammatory infiltrate, lobular inflammatory infiltrate, necrosis (clotting necrosis with pinkish cytoplasm and imprecise limits with pyknotic nuclei), microvesicular steatosis (clear droplets, well-delimited intracellular cells), detrabeculation (hepatocellular retraction with isolated hepatocytes, without forming beams), and sinusoidal cells (hypertrophied or hyperplastic). We then calculated the grade of the hepatic IR using the procedure of Quireze et al. (2006) [63]. The paraments were graded as follows: 0 points for absent, 1 point for mild, 2 points for moderate, and 3 points for severe. Congestion/bleeding received 2 points, and necrosis received 3 points. The grading accordingly varied from a minimum of 0 to a maximum of 30 points. The histological grades of congestion (congestion × 2) and necrosis (necrosis × 3) were also analysed separately.

### 4.6. Statistical Analyses

We present means and standard deviations for the variables considered to be parametric; we analysed those data using the unpaired Student’s *t*-test for two-by-two analysis. The non-parametric results were analysed using the Kruskal–Wallis test. The two-way ANOVA test was performed on the portal flow measures at the specified times. We used GraphPad Prism 9.0 (GraphPad Software, Boston, MA, USA); *p* < 0.05 was considered to be statistically significant.

## Figures and Tables

**Figure 1 ijms-25-08833-f001:**
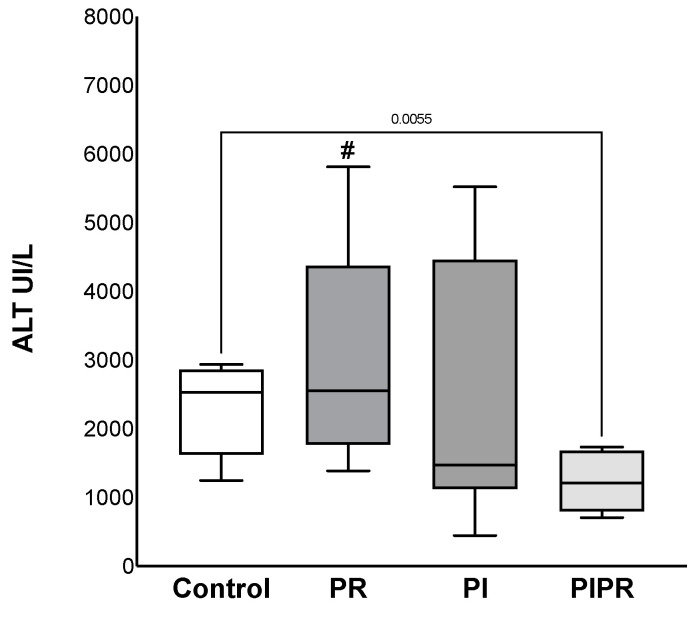
ALT dosages of the four groups: Control, treated with ascorbic acid before ischaemia (PI), treated before reperfusion (PR), and treated both before ischaemia and before reperfusion (PIPR). # *p* < 0.05 compared to PIPR. Statistical analysis was performed using unpaired Student’s *t*-test.

**Figure 2 ijms-25-08833-f002:**
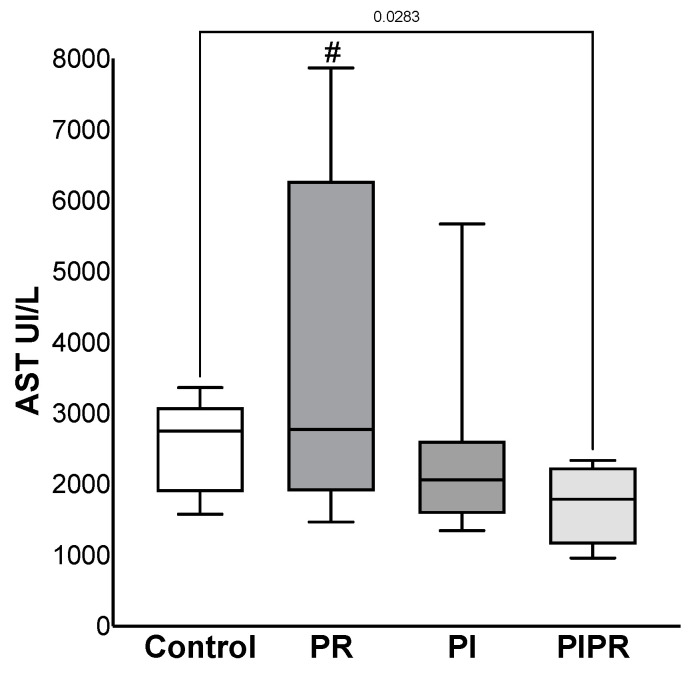
AST dosages of the four groups: Control, treated with ascorbic acid before ischaemia (PI), treated before reperfusion (PR), and treated both before ischaemia and before reperfusion (PIPR). # *p* < 0.05 compared to PIPR. Statistical analysis was performed using unpaired Student’s *t*-test.

**Figure 3 ijms-25-08833-f003:**
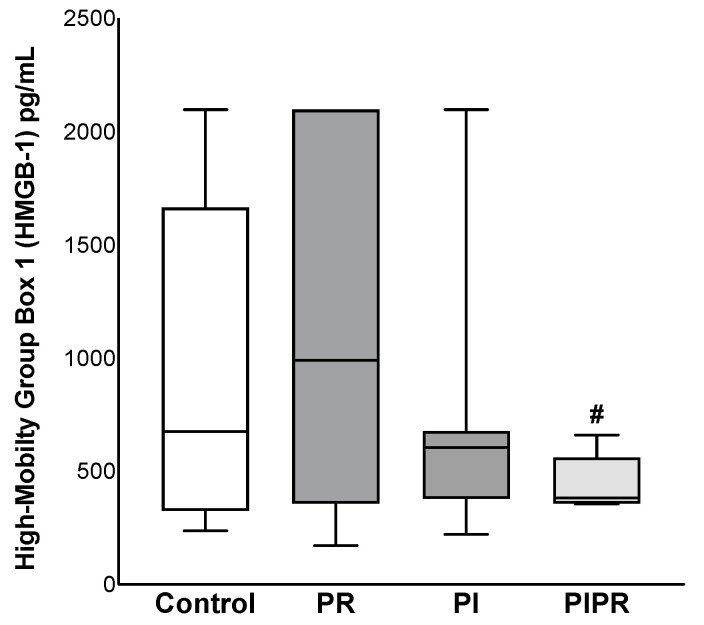
High-mobility group box-1 (HMGB-1) dosages of the four groups: Control, treated with ascorbic acid before ischaemia (PI), treated before reperfusion (PR), and treated both before ischaemia and before reperfusion (PIPR). # *p* < 0.05 compared to PR. Statistical analysis was performed using unpaired Student’s *t*-test.

**Figure 4 ijms-25-08833-f004:**
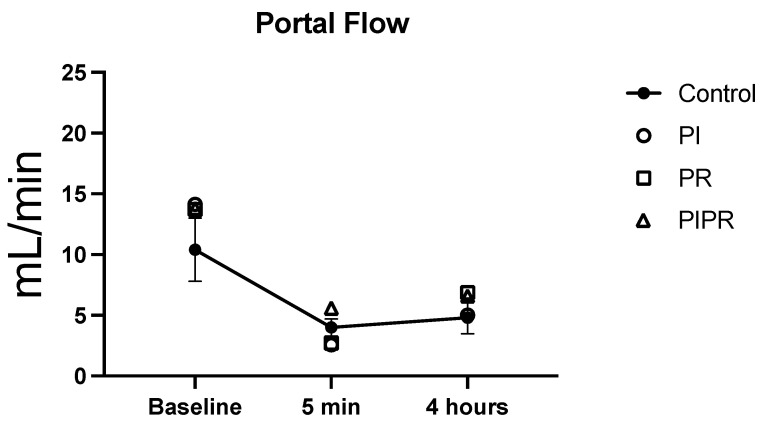
Portal flow at baseline, 5 min after reperfusion, and 4 h after reperfusion in the groups. Two-way ANOVA showed significant difference between times (*p* < 0.01) but not between the groups (*p* > 0.05).

**Figure 5 ijms-25-08833-f005:**
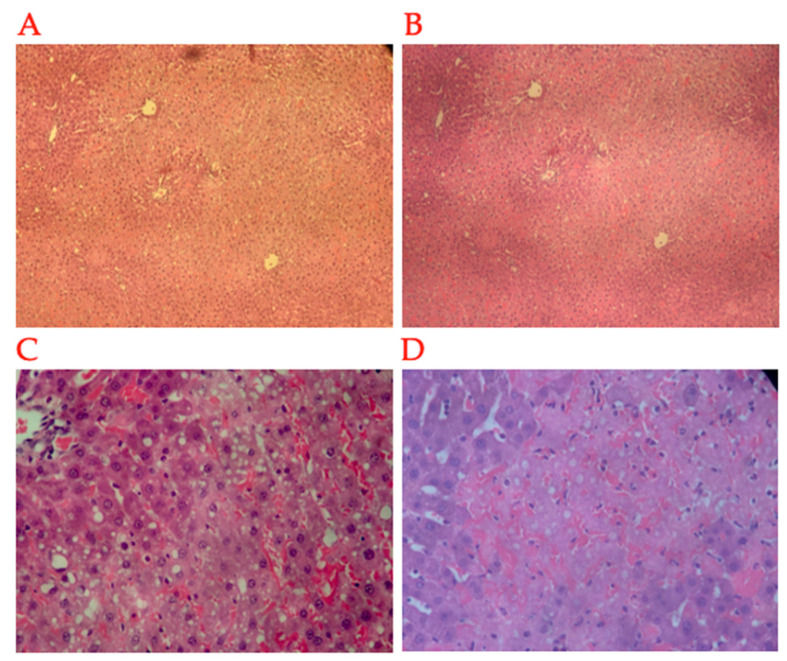
Liver sample in HE (haematoxylin–eosin) showing (**A**) grade II steatosis; (**B**) steatosis and zone III coagulative necrosis; (**C**) microvesicular steatosis and congestion; (**D**) coagulative necrosis and congestion.

**Figure 6 ijms-25-08833-f006:**
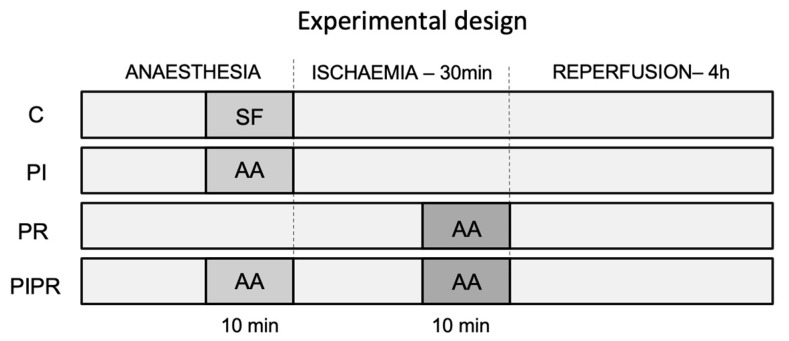
Study design. C, control group; PI, pre-ischaemia group; PR, pre-reperfusion group; PIPR, pre-ischaemia and pre-reperfusion group; SS, saline solution; AA, ascorbic acid.

**Table 1 ijms-25-08833-t001:** Biochemical results.

	Control	PI	PR	PIPR
pH	7.08 ± 0.07	7.19 ± 0.1	7.11 ± 0.13	7.11 ± 0.08
Base excess	−7.98 ± 2.39	−2.85 ± 6.18	−8.29 ± 4.88	−7.31 ± 5.21
HCO_3_	20.56 ± 1.75	24.40 ± 5.19	19.81 ± 3.46	19.81 ± 4.44
Haemoglobin (g/dL)	19.28 ± 0.59	17.73 ± 1.48	17.71 ± 1.40	17.71 ± 1.19
Potassium (mEq/L)	5.40 ± 0.76	6.14 ± 0.71	6.15 ± 0.69	6.15 ± 0.66
Ionised calcium (mg/dL)	3.93 ± 0.72	4.17 ± 0.83	3.48 ± 0.72	3.48 ± 0.91
Lactate (mg/dL)	31.00 ± 20.22	25.75 ± 10.73	34.50 ± 15.05	34.50 ± 9.50
Glucose (mg/dL)	327 ± 156	311 ± 105	340 ± 91	340 ± 212

**Table 2 ijms-25-08833-t002:** Interleukin levels.

	Control	PI	PR	PIPR
IL-1beta	412.46 ± 373.47	192.82 ± 136.06	711.10 ± 1085.13	414.39 ± 277.03
IL-6	544.12 ± 992.88	688.00 ± 1212.27	1225.11 ± 1314.19	903.42 ± 1262.83
IL-10	699.08 ± 647.46	401.97 ± 178.27	763.92 ± 513.45	601.60 ± 241.56
IL-12p70	165.93 ± 242.07	84.79 ± 86.25	85.20 ± 142.43	162.94 ± 180.11
TNF alfa	12.81 ± 14.47	12.63 ± 5.64	11.81 ± 7.40	12.82 ± 9.10
HMGB-1	930.61 ± 752.51	712.16 ± 591.13	1151.83 ± 849.92	444.19 ± 124.11

**Table 3 ijms-25-08833-t003:** Histopathology.

Variable	Total, *n* = 27	Control, *n* = 5	PI, *n* = 8	PR, *n* = 8	PIPR, *n* = 6
Congestion					
Absent	2 (7.4%)	1 (20%)	1 (12%)	0 (0%)	0 (0%)
Mild	16 (59%)	4 (80%)	3 (38%)	5 (62%)	4 (67%)
Moderate	7 (26%)	0 (0%)	4 (50%)	1 (12%)	2 (33%)
Severe	2 (7.4%)	0 (0%)	0 (0%)	2 (25%)	0 (0%)
Portal inflammatory infiltrate					
No	26 (96%)	5 (100%)	7 (88%)	8 (100%)	6 (100%)
Yes	1 (3.7%)	0 (0%)	1 (12%)	0 (0%)	0 (0%)
Lobular inflammatory infiltrate					
No	23 (85%)	5 (100%)	6 (75%)	6 (75%)	6 (100%)
Yes	4 (15%)	0 (0%)	2 (25%)	2 (25%)	0 (0%)
Detrabeculation					
No	19 (70%)	4 (80%)	6 (75%)	5 (62%)	4 (67%)
Yes	8 (30%)	1 (20%)	2 (25%)	3 (38%)	2 (33%)
Necrosis					
No	13 (48%)	3 (60%)	3 (38%)	4 (50%)	3 (50%)
Mild	5 (19%)	2 (40%)	1 (12%)	0 (0%)	2 (33%)
Moderate	7 (26%)	0 (0%)	4 (50%)	2 (25%)	1 (17%)
Severe	2 (7.4%)	0 (0%)	0 (0%)	2 (25%)	0 (0%)
Steatosis					
No	8 (30%)	2 (40%)	2 (25%)	1 (12%)	3 (50%)
Mild	14 (52%)	2 (40%)	6 (75%)	4 (50%)	2 (33%)
Moderate	5 (19%)	1 (20%)	0 (0%)	3 (38%)	1 (17%)
**Sinusoidal Cells**					
Mild	26 (96%)	5 (100%)	7 (88%)	8 (100%)	6 (100%)
Moderate	1 (3.7%)	0 (0%)	1 (12%)	0 (0%)	0 (0%)

## Data Availability

Data is contained within the article.

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
