# Peer review of "The Effect of Ascorbic Acid on Hepatic Ischaemia–Reperfusion Injury in Wistar Rats: An Experimental Study"

_ijms, 2024, doi:10.3390/ijms25168833_

Round 1

Reviewer 1 Report

Comments and Suggestions for Authors

In the manuscript titled “Effect of Ascorbic Acid on Hepatic Ischaemia-Reperfusion Injury in Wistar Rats: An Experimental Study,” this research investigates the protective effects of ascorbic acid against liver ischaemia-reperfusion (IR) injury, a condition that can lead to liver cell death through oxidative stress, immune cell activation, cytokine release, and damage-associated molecular patterns. The study evaluates ascorbic acid's effects on 36 male Wistar rats, divided into control and experimental groups, with the experimental groups receiving the same total dose of ascorbic acid at different time points: before ischaemia, before reperfusion, or before both. The results indicate that rats administered ascorbic acid before both ischaemia and reperfusion exhibit lower liver enzyme levels, reduced inflammation, and improved portal venous flow. This suggests that divided doses of ascorbic acid administered before IR may mitigate liver injury.

Additionally, I have some specific queries and comments regarding the manuscript:

In Line 58, there are two periods following “p=0.01” Could you please correct this?

Can you explain the abbreviations PI, PR, and PIPR used in the manuscript? Currently, these are only annotated in the figures, but a clear explanation within the text would be beneficial.

What does “XXX” refer to in Line 58?

In Table 1, please clarify what the symbols “-” and “`” signify in the context of base excess values.

Furthermore, as shown in Figures 1 to 3, the standard deviation is notably larger in the PI group compared to the PR, PIPR, and control groups for ALT dosages, AST dosages, and HMGB-1 dosage. Could the authors provide an explanation for this observed variability?

Lastly, what are the observed benefits of administering a divided dose of ascorbic acid (AA) before ischaemia and before reperfusion in the context of liver damage? How might this strategy be applied in clinical settings such as hepatic resections and transplants? An elaboration on these points would enhance the manuscript's relevance and applicability to clinical practice.

Comments on the Quality of English Language

The manuscript generally reads well, but there are a few instances where the punctuation could be improved.

Author Response

3. Point-by-point response to Comments and Suggestions for Authors

Comments and Suggestions for Authors

“In the manuscript titled “Effect of Ascorbic Acid on Hepatic Ischaemia-Reperfusion Injury in Wistar Rats: An Experimental Study,” this research investigates the protective effects of ascorbic acid against liver ischaemia-reperfusion (IR) injury, a condition that can lead to liver cell death through oxidative stress, immune cell activation, cytokine release, and damage-associated molecular patterns. The study evaluates ascorbic acid's effects on 36 male Wistar rats, divided into control and experimental groups, with the experimental groups receiving the same total dose of ascorbic acid at different time points: before ischaemia, before reperfusion, or before both. The results indicate that rats administered ascorbic acid before both ischaemia and reperfusion exhibit lower liver enzyme levels, reduced inflammation, and improved portal venous flow. This suggests that divided doses of ascorbic acid administered before IR may mitigate liver injury.

Additionally, I have some specific queries and comments regarding the manuscript:”

We sincerely appreciate the time and dedication you invested in reviewing our article. Your insightful comments and suggestions have significantly enriched our work and enhanced its presentation. Please find our detailed responses to each comment below.

Comment 1: In Line 58, there are two periods following “p=0.01” Could you please correct this?

Response 1: Corrected.

Comment 2: Can you explain the abbreviations PI, PR, and PIPR used in the manuscript? Currently, these are only annotated in the figures, but a clear explanation within the text would be beneficial.

Response 2: We explained the abrevviations in the text on lines 53-55, in this way “The C (control) group therefore contained five animals, the PIPR (pre-ischemia and pre-reperfusion) group contained six animals, and the PR (pre-reperfusion) and PI (pre-ischemia) groups”.

Comment 3: What does “XXX” refer to in Line 58?

Response 3: We corrected this error, the text now is “A similar effect was not observed when we compared PIPR with the PI group”

Comment 4: In Table 1, please clarify what the symbols “-” and “`” signify in the context of base excess values.

Response 4: We corrected this error in table 1, that symbols should not be there, they were erased.

Comment 5: Furthermore, as shown in Figures 1 to 3, the standard deviation is notably larger in the PI group compared to the PR, PIPR, and control groups for ALT dosages, AST dosages, and HMGB-1 dosage. Could the authors provide an explanation for this observed variability?

Response 5: About the standard deviation (SD) in the PI group, we emphasize that both PI and PR groups showed a little larger SD and the C e PIPR groups showed a more narrow SD. As pointed in the discussion (lines 259-267), the study does have limitations, including a relatively small sample size. However, given the experimental nature of the study, the well-defined subject profile, and the ethical constraints of animal research, the sample size was deemed appropriate. Small sample sizes can introduce greater variability due to chance or outlier effects, leading to larger standard deviations. As all experimental groups received the same drug, minimal differences in results were anticipated. Nonetheless, we observed statistically significant differences, suggesting that using additional animals would have been unnecessary. This may have influenced the observed standard deviation variability. We are confident that this does not compromise the quality of our research. Addressing this point in detail would likely lengthen the article unnecessarily and potentially detract from its objectivity.

Comment 6: Lastly, what are the observed benefits of administering a divided dose of ascorbic acid (AA) before ischaemia and before reperfusion in the context of liver damage? How might this strategy be applied in clinical settings such as hepatic resections and transplants? An elaboration on these points would enhance the manuscript's relevance and applicability to clinical practice.

Response 6: Agree. We have, accordingly, added the paragraphs between lines 206-225 to address these applicability concerns, especially indicated on the lines 222-225. To maintain brevity and objectivity, we have chosen not to delve further into these details within the text to emphasize this point. The text now is this “Extracellular levels of AA in hepatic tissues increase during ischaemia and during the beginning of reperfusion; a reduction then ensues in subsequent minutes [44]. That situation indicates the displacement of AA to ischaemic tissues and its ultimate consumption. Levels of AA in ischaemic hepatic lobes are also lower than those in non-ischaemic hepatic lobes [32]. That finding is likely due to the rapid consumption of AA. An infusion of AA before ischaemia sets up a stockpile, and a second dose before reperfusion can optimize the treatment. That second dose will have a more pronounced effect on reperfusion, and displacement of AA from the rest of the body to the affected area can begin.

The initial stages of IR injury-attenuating strategies can affect their ultimate effects; differences between pre-ischaemia and pre-reperfusion administration have already been observed [29, 38, 40, 45]. Here, we highlight the utility of administering effect a divided dose of AA: half before ischaemia and half before reperfusion. This situation can be observed in the expression of liver damage by transaminases; animals in the PIPR group—that is, the group that received half a dose of AA before ischaemia and the other half before reperfusion—exhibited the best profiles. Ascorbic acid is very water soluble; after parenteral infusion, it is distributed throughout tissues and has an immediate effect [24, 25, 28, 46-48]. This infusion strategy could be utilized during anaesthesia for both liver resections and transplants; in resections, one dose could be given at the start of surgery and another after periods of ischaemia; for transplants, doses could be administered before organ procurement and again prior to reperfusion.“

Comments on the Quality of English Language

Comment 7: The manuscript generally reads well, but there are a few instances where the punctuation could be improved

Response 7: We corrected some errors and submitted our text to a native English speaker review, trough the “American Manuscript Editors” for major review.

Reviewer 2 Report

Comments and Suggestions for Authors

In this study, authors evaluated the effect of ascorbic during hepatic IR injury in rats. Although, the roles of AA during IR/I are well studied, the infusion at different time points during hepatic IR seems to be the novelty approach of this study.

I suggest a major revision.

Line 236 and 263: what mean by XXX?

Line 277: Please explain the choice of AA dose and add a reference.

Line 303: please merge the two paragraph "Portal venous flow"

Line 343 and line 346: the authors don’t respect the journal template.

Authors should add image for the histopathology result (histology)

Author Response

3. Point-by-point response to Comments and Suggestions for Authors

Comments 1: In this study, authors evaluated the effect of ascorbic during hepatic IR injury in rats. Although, the roles of AA during IR/I are well studied, the infusion at different time points during hepatic IR seems to be the novelty approach of this study.

I suggest a major revision.

 We sincerely appreciate the time and dedication you invested in reviewing our article. Your insightful comments and suggestions have significantly enriched our work and enhanced its presentation. Please find our detailed responses to each comment below.

Comment 1: Line 236 and 263: what mean by XXX?

Response 1: Thank you for pointing this out, we corrected this error, the XXX was substituted by “HMGB-1”

Comment 2: Line 277: Please explain the choice of AA dose and add a reference.

Response 2: Thank you for pointing this out, we lines 291-293, as acordinly: “. The selected dose was determined based on existing pharmacokinetic data[41] and prior animal studies[16,17,19,20].“ We also added a study design in lines 294-297, as following:

Figure 6. Study design. C, control group; PI Pre-ischaemia group; PR, Pre-reperfusion group; PIPR, pre-ischameia and pre-reperfusion group; SS, Saline solution; AA, Ascorbic acid.

Comment 3: Line 303: please merge the two paragraph "Portal venous flow"

Response 3: Thank you for pointing this out, we corrected an error on this lines,, now its as following:

4.3 Portal venous flow

Portal venous flow was measured using a perivascular probe (PROBE NAME, Transonic Systems Inc., NY, USA) connected to a flow meter (TS420 Animal Research Flowmeter, Transonic Systems Inc.) We conducted those measurements three times during the procedure: before starting ischaemia (basal portal flow), 5 min after reperfusion, and 4 hours after reperfusion.

4.4 Laboratory analyses (liver enzymes, blood gas, interleukins and High-Mobility Group Box-1))

We evaluated transaminases, aspartate aminotransferase (AST), and alanine aminotransferase (ALT) 4 hours after reperfusion. We quantified AST and ALT (in IU/L) using the optimised ultraviolet method (COBAS MIRA, Roche Diagnostics, Rotkreuz, Switzerland). The results were expressed in units per litre. Glucose (mg/dL), lactate (mg/dL), potassium (mmol/L), calcium (mg/dL), pH, bicarbonate (BIC) (mmol/L), and haemoglobin (g/dL) were quantified using a gas analyser (ABL800 Flex; Radiometer Medical ApS, Brønshøj, Denmark).”

Comment 4: Line 343 and line 346: the authors don’t respect the journal template.

Response 4: Thank you for pointing this out, we corrected this error. Now this lines are as following: “using an atraumatic microvascular clamp (MiniClamp Bulldog dietrich, Prime Instrumentos Cirúrgicos, São Paulo, BR)” and “Graph Pad Prism 9.0 (GraphPad Software, Boston, USA); p<0.05 was considered to be stat”

Comment 5: Authors should add image for the histopathology result (histology)

Response 5: Thank you for pointing this out, we added histopathology images in figure 5, lines 162-167, as following:

A                           B

C                           D

Figure 5. Liver sample in HE (hematoxylin- eosin) showing A, Grade II steatosis; B, Steatosis and zone III coagulative necrosis; C, Microvesicular steatosis and congestion; D, Coagulative necrosis and congestion

4. Response to Comments on the Quality of English Language

Comment 6: Point 1: English language fine. No issues detected

Response 1: Thanks.

5. Additional clarifications

We would like to thanks for the suggestions made that improved our papers quality.

Reviewer 3 Report

Comments and Suggestions for Authors

Author (s) attempted to fine-tune the window of Ascorbic acid treatment during the Liver IR Pathologic states. I would like to appreciate author (s) for their tremendous efforts in conducting those experiments in mouse, which could impose several challenges during the study. I would like to suggest the author (s) to bring more clarity in the manuscript writing, several parts in the manuscripts lack background information or data are not clearly explained. Author (s) must appreciate that presented work should be acknowledged by a broad scientific community, it is critical that manuscript content must be comprehensive and clear.

Comments on the Quality of English Language

Writing parts certainly needs to be improved

Author Response

3. Point-by-point response to Comments and Suggestions for Authors

Comments 1: Author (s) attempted to fine-tune the window of Ascorbic acid treatment during the Liver IR Pathologic states. I would like to appreciate author (s) for their tremendous efforts in conducting those experiments in mouse, which could impose several challenges during the study. I would like to suggest the author (s) to bring more clarity in the manuscript writing, several parts in the manuscripts lack background information or data are not clearly explained. Author (s) must appreciate that presented work should be acknowledged by a broad scientific community, it is critical that manuscript content must be comprehensive and clear.

 We sincerely appreciate the time and dedication you invested in reviewing our article. Your insightful comments and suggestions have significantly enriched our work and enhanced its presentation. We have refined our writing and enhanced the presentation of our methods and results. Several changes were made, for example, to clarify key findings, such as a schematic overview of the study design and some representative histopatoly images:

Figure 6. Study design. C, control group; PI Pre-ischaemia group; PR, Pre-reperfusion group; PIPR, pre-ischameia and pre-reperfusion group; SS, Saline solution; AA, Ascorbic acid.

A                           B

C                           D

Figure 5. Liver sample in HE (hematoxylin- eosin) showing A, Grade II steatosis; B, Steatosis and zone III coagulative necrosis; C, Microvesicular steatosis and congestion; D, Coagulative necrosis and congestion

4. Response to Comments on the Quality of English Language

Comment 6: Writing parts certainly needs to be improved

Response 1: We corrected some errors and submitted our text to a native English speaker review, trough the “American Manuscript Editors” for major review.

5. Additional clarifications

We would like to thanks for the suggestions made that improved our papers quality.

Round 2

Reviewer 1 Report

Comments and Suggestions for Authors

The updated manuscript looks pretty good. The responses from the authors are acceptable and reasonable. I don't have any questions about this paper.

Reviewer 2 Report

Comments and Suggestions for Authors

In general, the manuscript is improved and the authors addressed all comments. Although, about the histology image, the authors should provide a histology for each animal groups (C, PI,PR and PIPR).

Reviewer 3 Report

Comments and Suggestions for Authors

Thanks for the changes in the manuscript.